# Learning Resilient Molecular Representations with Dynamic Multi-Modal Fusion

**Indra Priyadarsini S**
IBM Research - Tokyo
indra.ipd@ibm.com

**Seiji Takeda**
IBM Research - Tokyo
seijitkd@jp.ibm.com

**Lisa Hamada**
IBM Research - Tokyo
lisa.hamada@ibm.com

**Sina Klampt**
IBM Research - Tokyo
sina.klampt@ibm.com

**Takao Moriyama**
IBM Research - Tokyo
moriyama@jp.ibm.com

## Abstract

Recent advances in machine learning have transformed molecular property prediction, with large-scale representation models trained on diverse modalities such as SMILES, SELFIES, graph-based embeddings, etc. While multi-modal fusion offers richer insights than unimodal approaches, traditional fusion methods often assign static importance across modalities, leading to redundancy and poor robustness under missing-modality conditions. We introduce a Dynamic Multi-Modal Fusion framework, a self-supervised approach that adaptively integrates heterogeneous molecular embeddings. The framework employs intra-modal gating for feature selection, inter-modal attention for adaptive weighting, and cross-modal reconstruction to enforce information exchange across modalities. Training is guided by progressive modality masking, enabling the fused representation to remain informative even when some inputs are absent. Preliminary evaluations on the MoleculeNet benchmark demonstrate that our method improves reconstruction and modality alignment while achieving superior performance on downstream property prediction tasks compared to unimodal and naïve fusion baselines. These results highlight the importance of dynamic gating, entropy-regularized attention, and reconstruction-driven learning in building robust molecular fusion models.

## 1 Introduction

Molecular property prediction is central to drug discovery, materials science, and computational chemistry. Traditional cheminformatics relied heavily on handcrafted descriptors, but recent advances in deep learning have shifted the field towards learned molecular embeddings. String-based representations such as SMILES and SELFIES, along with graph-based models that encode molecular structures as graphs, have proven highly effective.These representation models are extensively used for tasks such as molecular property prediction, where their ability to capture and encode crucial molecular features has demonstrated remarkable efficacy (1; 2; 3; 4; 5; 6; 7; 8). These modalities, however, emphasize different facets of molecular structure: SMILES models capture sequential dependencies, SELFIES enforces chemical validity via a robust grammar, and graph encoders reflect topological organization. This complementarity motivates multi-modal fusion, an approach that has delivered state-of-the-art performance in vision, language, healthcare, and autonomous systems by leveraging signals from heterogeneous sources (9; 10; 11).

Realizing the benefits of multimodality in chemistry and material science, however, presents two persistent challenges. First, missing modalities are common: data pipelines are often incomplete and not all molecules have all representations due to computational cost or preprocessing failures. Second, noisy or redundant features arise within and across modalities, which can dilute signal. Conventional

fusion strategies such as naive concatenation, averaging, or simple pooling—presume complete inputs, treat modality-specific noise uniformly, and cannot adjust per-molecule contributions, leading to brittle performance when data are sparse, imbalanced or missing.

To address these challenges, in this work, we propose a self-supervised dynamic fusion framework that learns a single, compact molecular embedding by dynamically selecting informative features, adapting modality weights on a per-sample basis, and reconstructing missing embeddings during training. By incorporating a curriculum of progressive modality masking, the proposed dynamic fusion approach is trained to remain robust under missing-modality conditions, making it particularly useful in real-world molecular applications.

## 2  Proposed Dynamic Fusion Framework

In this section, we outline the methodological framework of our proposed approach. Fig. 1 illustrates the schematic of the proposed dynamic fusion approach. The core objective of our dynamic multimodal fusion model is to enhance robustness and performance by adaptively tailoring the fusion process to inputs from distinct unimodal models and efficient handling of missing or scarce paired data.

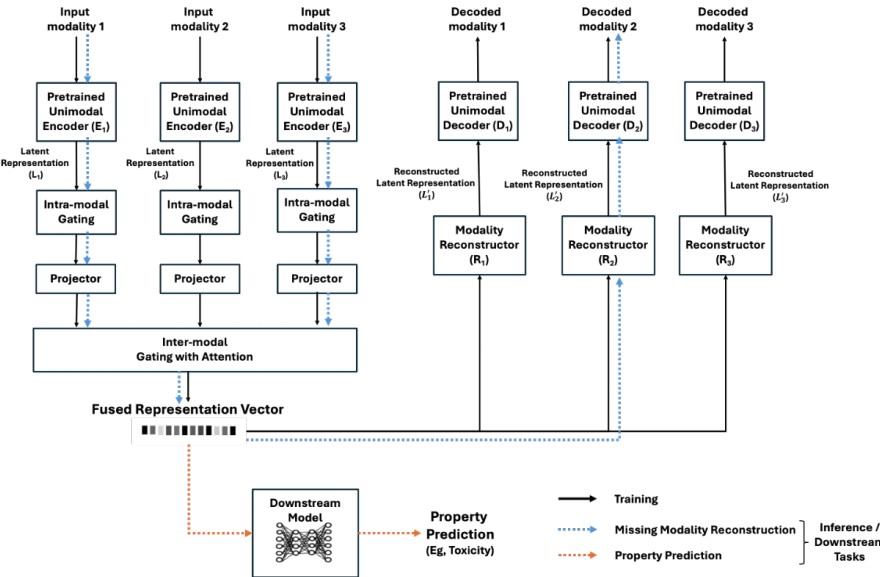

Figure 1: Block diagram of the proposed dynamic fusion model

We assume that for each molecule, multiple pretrained embeddings are available by means of their corresponding unimodal models: for example, SMILES-based transformer embeddings (768 dimensions) (12), SELFIES-based embeddings (1024 dimensions)(13), and graph neural network embeddings from MHG-GED (1024 dimensions) (14). Each embedding is treated as a modality. The task is to produce a fused embedding of fixed dimension (512 in our experiments) that integrates information across modalities and generalizes well to downstream supervised tasks.

The framework first applies intra-modal gating networks, small MLPs that output sigmoid weights to select informative features within each modality. The gated embeddings are then linearly projected into a shared latent space, enabling direct comparison across heterogeneous dimensions. Fusion is achieved through an inter-modal attention mechanism, which assigns adaptive weights to modalities per sample. Missing inputs are masked during softmax normalization, ensuring that absent modalities do not influence the fused representation. The final fused embedding is a weighted sum of projected embeddings. To improve robustness, our framework incorporates cross-modal reconstruction: decoder networks attempt to reconstruct masked embeddings from the fused representation. Training randomly masks modalities and requires their reconstruction, enforcing redundancy and making the fused embedding informative even with incomplete inputs.

The proposed dynamic fusion framework is trained with a multi-component objective function comprising of (i) reconstruction loss ensures recovery of masked modalities; (ii) alignment loss keeps the fused embedding consistent with modality projections; (iii) diversity loss reduces redundancy between gating patterns; (iv) entropy regularization prevents attention collapse; and (v) sparsity regularization promotes compact feature selection.

$$\mathcal{L} = \mathcal{L}_{rec} + \lambda_a \mathcal{L}_{align} + \lambda_d \mathcal{L}_{div} + \lambda_e \mathcal{L}_{ent} + \lambda_s \mathcal{L}_{sparse}. \tag{1}$$

$$\mathcal{L}_{rec} = \sum_{m=1}^{M} \|D_m(f) - x_m\|_2^2, \quad \text{for masked modalities only.} \tag{2}$$

$$\mathcal{L}_{align} = \frac{1}{M} \sum_{m=1}^{M} \text{softplus}\big(-\cos(f, z_m)\big). \tag{3}$$

$$\mathcal{L}_{div} = \frac{1}{\binom{M}{2}} \sum_{i<j} \cos(g_i, g_j)^2. \tag{4}$$

$$\mathcal{L}_{ent} = -\mathbb{E}\left[\sum_{m=1}^{M} \alpha_m \log \alpha_m\right]. \tag{5}$$

$$\mathcal{L}_{sparse} = \frac{1}{M} \sum_{m=1}^{M} \|g_m(x_m)\|_1. \tag{6}$$

A key aspect of the training strategy is employing progressive masking, by gradually increasing the masking probability during training. This curriculum forces the model to handle increasingly difficult reconstruction tasks, leading to fused embeddings that are accurate, balanced, and resilient to missing modalities.

At inference, the fused embedding can be directly employed for downstream property prediction tasks or for reconstructing missing modalities, making the framework broadly applicable across diverse molecular learning scenarios.

## 3 Results and Discussions

To evaluate our proposed dynamic multimodal fusion approach, we consider three modalities : SMILES, SELFIES, and molecular graphs. Each modality's latent representation is derived using pretrained open-source models, ensuring a robust and scalable feature extraction process. For the SMILES modality, we utilize the encoder of the SMI-TED foundation model (6). This large-scale, open-source encoder-decoder model was pre-trained on a meticulously curated dataset of 91 million SMILES samples from PubChem, encompassing a total of 4 billion molecular tokens. For the SELFIES modality, we employ the SELFIES-TED model (13), an encoder-decoder architecture based on BART. This model was trained on molecular representations using the ZINC-22 (15) and PubChem (16) datasets, ensuring effective encoding of SELFIES representations. For the molecular graph modality, we leverage the MHG-GED model (14), an autoencoder that integrates Graph Neural Networks (GNNs) with Molecular Hypergraph Grammar (MHG), originally introduced in MHG-VAE (17). MHG-GNN encodes molecular structures as graphs, employing a Graph Isomorphism Network (GIN) that incorporates edge information to generate meaningful latent embeddings. To simulate real-world scenarios with missing data, modality-specific embeddings were randomly omitted during training phases.

As a preliminary analysis, we evaluate the performance of our proposed fusion method across six classification tasks from the MoleculeNet dataset. The evaluation includes comparisons between the respective unimodal, multimodal by naive concatenation and our proposed dynamic fusion method. The results are summarized in Table 1. As observed, multimodal by naïve concatenation generally outperforms unimodal approaches. However, its performance varies significantly based on the combination of modalities, and as the number of modalities increases, so does the computational overhead associated with identifying optimal modality combinations. Additionally, naïve concatenation leads to increased feature dimensionality, which can introduce redundancy and inefficiency. In contrast, our dynamic fusion approach surpasses unimodal cases and is competitive to naïve concatenation methods in majority of the tasks. By incorporating intra- and inter-modal gating mechanisms, our

Table 1: ROC-AUC on classification benchmarks (bold = best per column).

| | bace | bbbp | clintox | hiv | tox21 | sider |
|---|---|---|---|---|---|---|
| SMILES (SMI-TED) | 0.863 | 0.907 | 0.907 | 0.789 | 0.768 | 0.665 |
| SELFIES (SELFIES-TED) | 0.862 | 0.940 | 0.878 | 0.797 | 0.717 | 0.642 |
| Graph (MHG-GED) | 0.860 | 0.927 | 0.839 | 0.820 | 0.770 | 0.666 |
| SMILES ⊕ SELFIES | 0.855 | 0.952 | **0.921** | 0.811 | 0.759 | **0.674** |
| SELFIES ⊕ Graph | 0.882 | 0.949 | 0.904 | 0.804 | **0.792** | 0.667 |
| SMILES ⊕ Graph | **0.887** | 0.915 | 0.906 | **0.824** | 0.760 | 0.672 |
| SMILES ⊕ SELFIES ⊕ Graph | 0.862 | **0.954** | 0.917 | 0.814 | 0.770 | 0.670 |
| Dynamic Fusion | 0.875 | 0.939 | 0.895 | 0.815 | 0.749 | 0.673 |

approach adaptively selects and fuses the most informative features while effectively handling missing modalities. Furthermore, unlike naïve concatenation, which requires paired data for training, our method remains robust even in scenarios where certain modalities are absent, making it a more flexible and scalable solution for multimodal molecular representation learning.

To further evaluate the robustness of the proposed framework, we investigated performance degradation under missing-modality conditions at inference (Figure 2). Unlike naïve concatenation, which fails under incomplete inputs, dynamic fusion degrades gracefully as modalities are removed. The removal of a single modality generally caused only marginal drops in performance, and in some cases accuracy even improved, suggesting that the gating and attention modules suppress noisy or less informative inputs. Performance declines were more pronounced when multiple modalities were simultaneously masked, yet the model avoided catastrophic collapse, maintaining usable accuracy even under severe input degradation. The "all missing" condition, with up to 25% of inputs randomly removed, further demonstrated the framework's resilience, retaining predictive capacity despite substantial loss of modality information. To better understand the underlying mechanism, we analyzed the behavior of the inter-modal attention weights ($\alpha$) (Figure 5) and the effective of masking (Figure 4) on one of the datasets. On average, SMILES and MHG receive the highest attention scores ($\alpha \approx 0.5$ and $\alpha \approx 0.45$, respectively), while SELFIES contributes minimally. This indicates that the model identifies SMILES and MHG as more informative and stable modalities for the corresponding dataset. However, when progressively increasing the mask rate, the attention distribution shifts: reliance on SMILES and MHG decreases, while SELFIES gradually gains weight. At high mask rates, the contributions across modalities become more balanced, illustrating how the model dynamically reallocates attention to compensate for missing modalities. Together, these findings provide a consistent picture: the graceful degradation observed in Figure 2 is a direct result of the adaptive attention mechanism illustrated in Figure 4. The fusion model prioritizes strong modalities when available but can flexibly redistribute weights to weaker modalities under data scarcity, thereby reducing redundancy, avoiding modality collapse, and ensuring resilience in real-world scenarios where missing or noisy data are common.

Finally, we evaluated reconstruction quality using cosine similarity (Figure 3). The results demonstrate that the fused embedding retains sufficient information to recover missing modalities with high fidelity, achieving similarity scores above 0.9 in most cases. This reinforces the effectiveness of cross-modal reconstruction in maintaining alignment across modalities and providing resilience under incomplete data conditions.

## 4  Conclusion

In this work, we introduced a multimodal dynamic fusion framework with intra-modal gating, inter-modal attention, and multi-objective self-supervised learning. The proposed framework achieves superior robustness to missing modalities and improved downstream performance, while offering interpretability via gates and attention weights. The framework generalizes beyond molecules to other multimodal domains such as vision, language, and audio. Future works include extensive study on other tasks and interpretability.

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
