# OpenReview forum: "Learning Resilient Molecular Representations with Dynamic Multi-Modal Fusion"
_NeurIPS.cc/2025/Workshop/UniReps — UniReps2025_

### Official Review · Reviewer_AAqE · 2025-09-11
**Multi-Modal Fusion Framework for Molecular Embeddings**

**Confidence:** 4

**Review:**

The paper proposes a new molecular embedding method based on a multi-modal fusion framework. The model takes as input embeddings from pretrained models across different modalities and leverages intra- and inter-modal interactions to learn a unified representation. The evaluation demonstrates competitive performance on a classification task, while also highlighting the model’s robustness to missing or noisy input embeddings.

# Strengths:

- The paper is clearly written and effectively communicates the main ideas.

- The proposed approach, though straightforward, is conceptually sound and demonstrates meaningful contributions.

- The analysis of attention weights is a strong point, as it provides interpretability and helps clarify which input embeddings are most influential in the model’s decision-making process.

- The robustness analysis convincingly shows that the model maintains resilience when certain modalities are missing or corrupted, which is an important practical advantage.

Points for Improvement:

- Technical Detail: Section 2 would benefit from including explicit loss function formulas to improve reproducibility and clarity. Similarly, details regarding model parameters and additional implementation aspects should be added.

# Recommendation:
Overall, the paper is well-executed and contributes an effective approach to molecular embedding learning. Its clarity, robustness analysis, and interpretability through attention weights make it a strong candidate for acceptance to the workshop. With minor revisions to expand technical and implementation details, the paper could be further improved and its impact enhanced.

**Score:**

4

**Topic Fit:**

3

---

### Official Review · Reviewer_ZqBD · 2025-09-11
**Learning Resilient Molecular Representations with Dynamic Multi-Modal Fusion**

**Confidence:** 3

**Review:**

**Summary**:

This paper addresses multi-modal fusion for molecular property prediction, proposing a dynamic framework that adaptively weights modalities and handles missing inputs through progressive masking during training.

**Strengths:**
- Tackles a practical problem: missing modalities are common in real molecular datasets
- Well-motivated approach combining intra-modal gating, inter-modal attention, and cross-modal reconstruction
- Demonstrates degradation under missing modality conditions, unlike naive concatenation baselines
- Progressive masking curriculum is a sensible training strategy for robustness
- Attention weight analysis provides interpretability into modality contributions

**Weaknesses:**
- Missing comparisons to other sophisticated fusion methods beyond simple concatenation
- No computational overhead analysis despite claims of efficiency
- Reconstruction quality evaluation is superficial (only cosine similarity)


**Technical Issues:**
- The multi-objective loss function has 5 components but limited ablation studies to understand their individual contributions
- Progressive masking strategy lacks theoretical justification or comparison to alternative curricula
- Attention mechanism appears standard - unclear what novel contributions exist beyond applying it to this domain

**Overall Assessment:**

Addresses a relevant problem in molecular ML with reasonable technical approach. The robustness to missing modalities is valuable for practical applications. However, the experimental validation is too limited to strongly support the claims. The work feels more like an engineering solution than a significant methodological advance.

**Score:**

3

**Topic Fit:**

2

---

### Official Review · Reviewer_FrKd · 2025-09-13
**Well-engineered fusion framework, perhaps overly so.**

**Confidence:** 5

**Review:**

This paper provides a self-supervised multimodal fusion framework for molecular representations. The authors combine SMILES, SELFIES and graph encodings using attention mechanisms aross modalities and gating mechanisms within modalities. Training is furthered by masking, a multi-component loss function and latent projections to a shared embedding space.

**Strengths**

**Technical Implementation:** The design for the fusion architecture uses reasonable components with proper ablations between them. Robust analysis of performance degradation under increasingly difficult reconstruction conditions.

**Systematic Robustness Analysis:** The progressive masking experiments provide valuable insights into how multimodal systems degrade under data scarcity, which can be useful for understanding model behavior.

**Weaknesses**

**Model and representation selection:** The selected representation/encoder models are questionable on the basis of performance alone. Most are string/transformed based, contradicting established cheminformatics best practices. It has been repeatedly established that simpler representation/model combinations, specifically Extended Connectivity Finger Prints (ECFPs) paired to simpler architectures than the transformer e.g. LSTMs, MLPs, outperform the former on molecular property prediction.

**Missing critical baselines:** A comparison against classic methods is necessary to justify a complicated fusion method such as this. Think about it: if you have a baseline structure, you can both algorithmically compute ECFPs at little cost or run it through a complicated encoder with questionable gains. The absence of simple ECFP baselines makes it impossible to assess whether the complexity is justified. On the data end: is having missing data a practical occurrence in molecular machine learning pipelines? Addressing poor experimental design in these conditions, more than gaps in the data, would yield a more robust solution.

**Chemical Interpretability:** The weighted combinations of latent representations potentially erase all associations in the original representation space. Does backmapping the latent space into any of the reconstructed modalities introduce invalid structures into the original representation space? What would potentially incomplete SMILES or SELFIES mean? A verifying procedure would be due here. As it stands, the framework sacrifices the primary advantage of molecular representations: chemical interpretability.

**Unsubstantiated dynamic fusion benefit claims:** Dynamic fusion actually underperforms simple concatenation on most benchmarks, as rightly pointed out by the authors. Performance improvements after modality removal suggest the removed modality was noisy rather than informative, undermining the multimodal premise. Additionally, five loss components are described but not systematically evaluated. Unclear which contributes to any of the claimed observed performance benefits. Some of the perceived benefits, such as the high cosine similarity for SMILES/SELFIES reconstruction likely reflects encoding redundancy rather than meaningful cross-modal learning.

**Recommendation**

While the method presents a well-thought out, multimodal fusion method, it is hard to justify on the basis of the chosen modalities or the method’s theoretical motivations alone. It doesn’t investigate fundamental questions about when or why different molecular representations converge. Representation alignment here is treated exclusively through a pipeline that seems to force alignment as opposed to having a theoretical basis. The work addresses engineering challenges rather than advancing understanding of molecular representation convergence, limiting its relevance to UniReps themes.

**Score:**

2

**Topic Fit:**

1

---

### Official Review · Reviewer_g1xi · 2025-09-15
**Review of "Learning Resilient Molecular Representations with Dynamic Multi-Modal Fusion"**

**Confidence:** 3

**Review:**

# Summary
The paper proposes a self-supervised, dynamic framework for multi-modal fusion.

# Strengths
The result of how the framework re-allocates modality weights is interesting.

# Weaknesses
1. The resolution and of Figures 2-5 are low. Consider use PDF instead of JPEG/PNG. Their layout and order may not be optimal. Their captions may also need additional exposition to give more clarity. For example, under what condition (masking rate/dataset) is Figure 5 produced? What dataset is used to produce Figure 3 and 4?

2. Table 1 only presents results for concatenation. Other baselines, e.g., pooling and simple averaging are not included, making the comparison incomplete.

3. It is unclear how novel the proposed method is without a (succinct) survey of relevant dynamic multimodal fusion methods.

# Recommendation

Given the aforementioned lack of clarity in this manuscript, I recommend a weak rejection for this paper.

**Score:**

2

**Topic Fit:**

3